# Estimating the effects of non-pharmaceutical interventions on the number of new infections with COVID-19 during the first epidemic wave

**Nicolas Banholzer**[1]*, **Eva van Weenen**[1], **Adrian Lison**[1], **Alberto Cenedese**[1], **Arne Seeliger**[1], **Bernhard Kratzwald**[1], **Daniel Tschernutter**[1], **Joan Puig Salles**[1], **Pierluigi Bottrighi**[1], **Sonja Lehtinen**[2], **Stefan Feuerriegel**[1], **Werner Vach**[3,4]

**1** Department of Management, Technology, and Economics, ETH Zurich, Zurich, Switzerland, **2** Department of Environmental Systems Science, ETH Zurich, Zurich, Switzerland, **3** Basel Academy for Quality and Research in Medicine, Basel, Switzerland, **4** Department of Environmental Sciences, University of Basel, Basel, Switzerland

* nbanholzer@ethz.ch

**Data Availability Statement:** We collected data from publicly available data sources (Johns Hopkins Coronavirus Resource Center for

## Abstract

The novel coronavirus (SARS-CoV-2) has rapidly developed into a global epidemic. To control its spread, countries have implemented non-pharmaceutical interventions (NPIs), such as school closures, bans of small gatherings, or even stay-at-home orders. Here we study the effectiveness of seven NPIs in reducing the number of new infections, which was inferred from the reported cases of COVID-19 using a semi-mechanistic Bayesian hierarchical model. Based on data from the first epidemic wave of $n = 20$ countries (i.e., the United States, Canada, Australia, the EU-15 countries, Norway, and Switzerland), we estimate the relative reduction in the number of new infections attributed to each NPI. Among the NPIs considered, bans of large gatherings were most effective, followed by venue and school closures, whereas stay-at-home orders and work-from-home orders were least effective. With this retrospective cross-country analysis, we provide estimates regarding the effectiveness of different NPIs during the first epidemic wave.

## 1 Introduction

The novel coronavirus (SARS-CoV-2) has developed into a global epidemic. Efforts to control the spread of SARS-CoV-2 focused on non-pharmaceutical interventions (NPIs). These represent public health-policy measures that were intended to diminish transmission rates and, to this end, aimed at reducing person-to-person contacts via so-called social distancing [1]. Examples include school closures, venue closures, or stay-at-home orders.

Early studies on the population-level effects of NPIs analyzed their effectiveness mostly within a single country [2–9]. Thereby, NPIs were often packaged into bundles and their combined effectiveness was assessed and confirmed. By combining data from multiple countries, a couple of studies have attempted to compare the effectiveness of individual NPIs [10–15];

epidemiological data; news reports and government resources for policy measures). All the public health information that we used is documented in the main text, the extended data, and supplementary tables. A preprocessed data file together with reproducible code is available from https://github.com/nbanho/npi_effectiveness_first_wave.

**Funding:** NB, EvW and SF acknowledge funding from the Swiss National Science Foundation (SNSF) as part of the Eccellenza grant 186932 on "Data-driven health management". The funding bodies had no control over design, conduct, data, analysis, review, reporting, or interpretation of the research conducted.

**Competing interests:** SF reports membership in a COVID-19 working group of the World Health Organization but without competing interest. JPS declares part-time employment at Luciole Medical outside of the submitted work. SF reports grants from the Swiss National Science Foundation outside of the submitted work. All other authors declare no competing interests. All competing interests do not alter our adherence to PLOS ONE policies on sharing data and materials.

however, the evidence from these studies regarding which NPIs were particularly effective is still inconclusive.

Here we contribute further evidence on the combined and individual effectiveness of NPIs. Using a semi-mechanistic Bayesian hierarchical model, we estimated the effects of NPIs on the number of new infections across $n = 20$ Western countries during the first epidemic wave: the United States, Canada, Australia, the EU-15 countries (Austria, Belgium, Denmark, Finland, France, Germany, Greece, Ireland, Italy, Luxembourg, the Netherlands, Portugal, Spain, Sweden, and the United Kingdom), Norway, and Switzerland. This amounts to $\sim 3.3$ million reported cases of coronavirus disease 2019 (COVID-19) and covers a population of $\sim 0.8$ billion people.

## 2 Methods

### 2.1 Data

Reported SARS-CoV-2 cases for each country between February and May 2020 were obtained from the Johns Hopkins Coronavirus Resource Center, which was developed for real-time tracking of reported cases of COVID-19 and directly aggregates cases recorded by local authorities in order to overcome time delays from alternative reporting bodies [16]. Hence, these numbers are supposed to account for all COVID-19 cases identified on a specific day.

Data on NPIs were collected by our research team. Their implementation dates were systematically obtained from government resources and news outlets before the NPIs were being classified into seven categories, based on definitions applicable across our sample of countries (Table 1): (1) bans of large gatherings, (2) school closures, (3) venue closures (e.g., shops, bars, restaurants, and venues for other recreational activities), (4) bans of small gatherings, (5) border closures, (6) stay-at-home orders prohibiting public movements without valid reason, and (7) work-from-home orders. Note that stay-at-home orders always implied bans of gatherings and venue closures, and bans of large gatherings implied bans of small gatherings. That is, for instance, if a country implemented a ban of small gatherings without yet having implemented a ban of large gatherings, then the implementation date for the ban of small and the ban of large gatherings is the same.

The implementation date of an NPI refers to the first day a measure went into action. The implementation dates were thoroughly checked to ensure correctness and consistency across countries. Overall, eight authors were involved in collecting, categorizing, and checking the data. Furthermore, local residents and/or native speakers were recruited from some countries

**Table 1. List of non-pharmaceutical interventions (NPIs).**

| NPI | Definition |
|---|---|
| Ban of large gatherings | Ban of public or private gatherings involving more than 50 people |
| School closure | Closure of schools (for primary schools) |
| Venue closure | Full-day closure of venues (i.e., the closure of some or all walk-in non-essential businesses like bars and restaurants, shops, and recreational facilities) |
| Border closure | Closure of national borders for individuals |
| Ban of small gatherings | Ban of public or private gatherings involving less than 50 people |
| Stay-at-home order | Prohibition of movement without valid reason (e.g., restricting mobility except to/from work, local supermarkets, and pharmacies) |
| Work-from-home order | Mandatory order to work from home (i.e., mostly related to office workers) if it is not essential to continue working at the workplace (as is mostly the case for, e.g., factories, laboratories, supermarkets, and pharmacies) |

in order to verify our encoding in cases where the interpretation of legal terms was ambiguous or where it was difficult to distinguish, for instance, whether an NPI was enforced or recommended. Sources and details on data collection for NPIs are provided in Section 8 in S1 Appendix.

Our selection of Western countries defined a sample that followed a similar and comparable overall strategy in controlling the COVID-19 outbreak. On the one hand, the national (and subnational) strategies consisted of similar NPIs, which we can expect to work in a similar manner, despite cultural and organisational differences between countries. On the other hand, the national (and subnational) strategies differed in the choice, timing, and sequencing of NPIs. Taken together, the setting of this study resembles that of a natural experiment, which allows us to learn about the effects of different NPIs. However, in countries with a federal structure, the timing of NPIs may differ between regions (e.g., states or territories). In countries with such regional variation, NPI data was collected at the regional level and, similar to Hsiang et al. [14], we took into account the cumulative share of the country's population that is affected by an NPI. Fig 1 summarizes our NPI data by comparing choice, timing and sequencing of NPIs across countries after considering regional variation within countries.

## 2.2 Model summary

In this section, we provide a short description of our model. A detailed description, including all modeling and prior choices is given in Section 1 in S1 Appendix.

Fig 2 provides a visual summary of our model structure. Our model links two unobserved quantities (i.e., the daily number of contagious subjects and the daily number of new infections) to an observed quantity (i.e., the number of reported new cases). The links consist of three components: (1) a regression type model relating the number of new infections to the number of contagious subjects, the country-specific daily transmission rate, and the presence of active measures; (2) a link between the number of new infections to the number of reported new cases; and (3) a link between the number of new infections and the number of contagious subjects.

The fundamental part of the first component is a model for the expected number $\mu$ of new infections $I_{jt}$ in country $j$ at day $t$. In the absence of any measure, this would be $\mu^{I_{jt}} = C_{jt} \, \delta_j$, where $C_{jt}$ are the number of contagious subjects and $\delta_j$ is the country-specific daily transmission rate. In the presence of NPIs, we multiply this with the reduction due to avoided infections, resulting in

$$\mu^{I_{jt}} = C_{jt} \; \delta_j \cdot \prod_{m=1}^{M}(1 - a_{mjt}), \tag{1}$$

where $a_{mjt}$ denotes the fraction of avoided infections due to NPI $m = 1, \ldots, M$ in country $j$ at day $t$. In case NPI $m$ is implemented and fully effective, $a_{mjt}$ is set equal to the value $\theta_m$. However, an NPI may not be fully effective in a country due to regional differences or because it may take a few days until subjects respond to the new measures. Hence, the general structure of $a_{mjt}$ is

$$a_{mjt} = \theta_m \sum_{r=1}^{R_j} p_{rj} f(T_{mrjt}), \tag{2}$$

where $p_{rj}$ is region's $r = 1, \ldots, R_j$ proportion of the total population in country $j$, where $T_{mrjt}$ denotes the number of days since the implementation of NPI $m$ in region $r$ of country $j$ at time $t$ (such that $T_{mrjt} = 1$ denotes the first day at which a reduction in the number of new infections

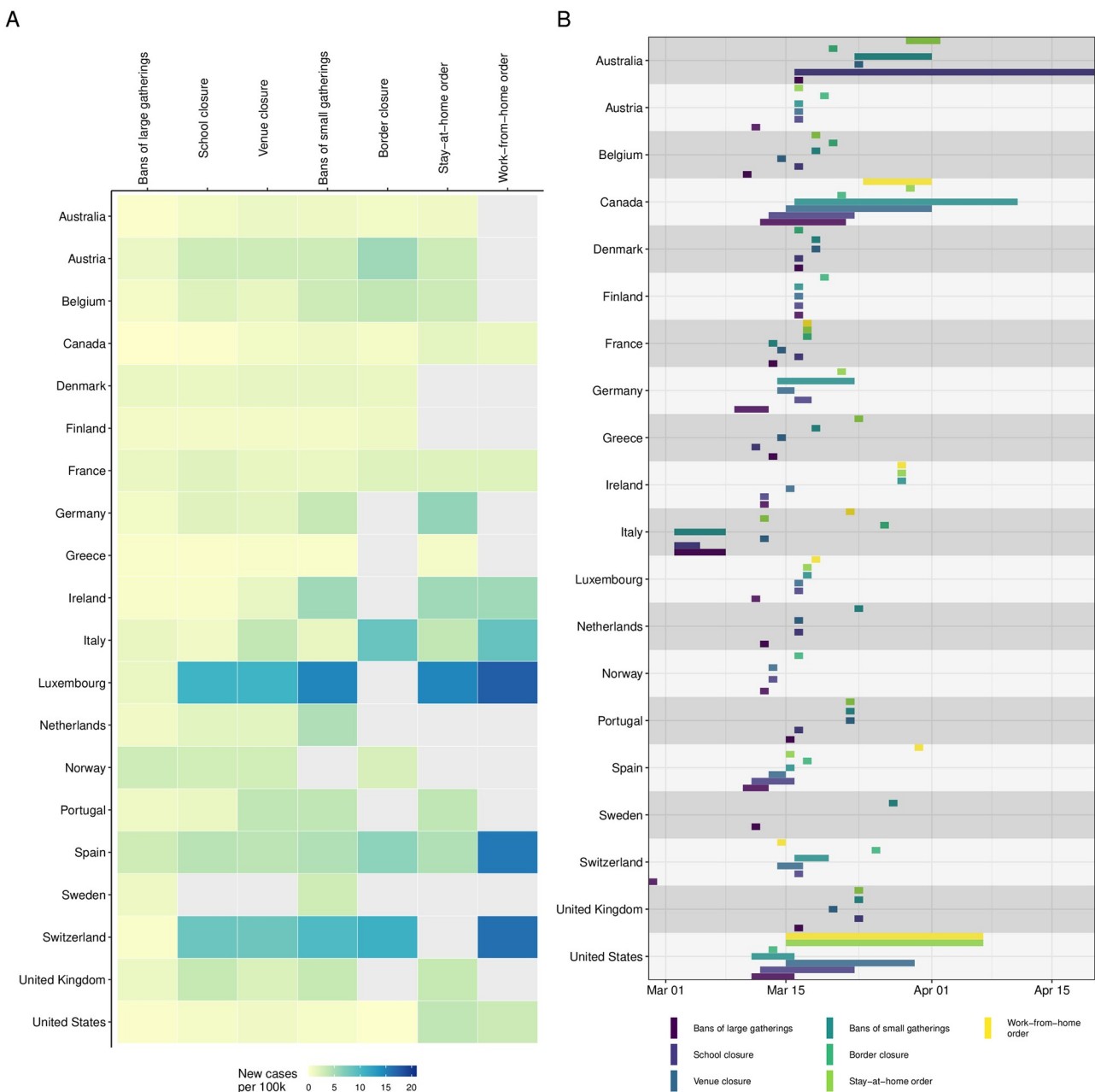

**Fig 1. Timing of NPIs by country.** (A) Number of new cases per 100,000 (rolling 7-day mean) when NPIs were first implemented across countries. For countries with regional variation in the implementation of NPIs, the number of new cases was averaged across regions. (B) Timeline of the implementation of NPIs. The horizontal lines show the time period in which NPIs were implemented within each country's regions. For most countries, there was no regional variation and the NPIs were implemented at one day across the entire country.

could be expected), and where $f(T_{mrjt})$ is a time-delayed response function, which is specified such that the response to an NPI increases from zero on day $T_{mrjt} = 0$ to one on day $T_{mrjt} = 3$ (Fig 3A), reflecting our expectation that NPIs typically require a few days until they are fully effective. The choice for the time-delayed response is varied as part of the sensitivity analysis.

The effect of an NPI when fully implemented is equal to $\theta_m$. Within our Bayesian framework, a mixture prior was used to model the effect $\theta_m$. The prior consists of a half-normal

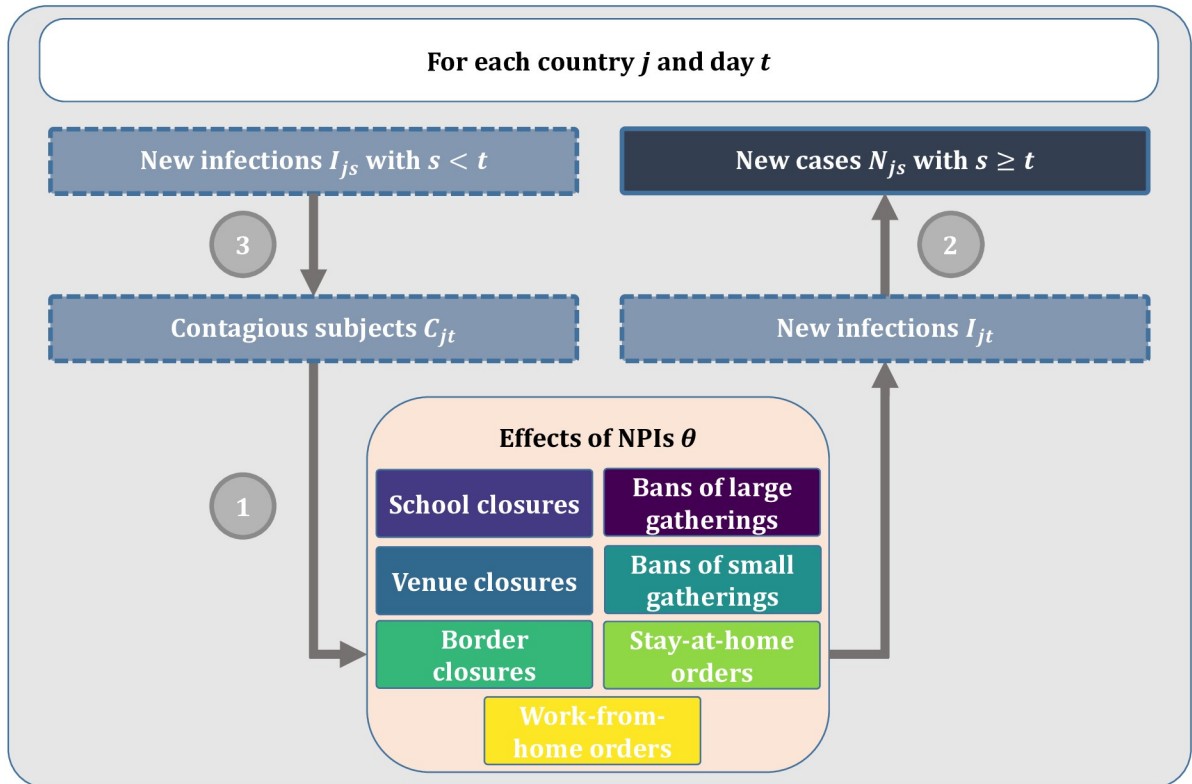

**Fig 2. Visual summary of the model structure.** (1) the number of new infections is modelled as a function of the number of contagious subjects, the country-specific daily transmission rate, and the reductions from active NPIs; (2) the observed number of new cases is a weighted sum of the number of new infections in the previous days; and (3) the number of contagious subjects is a weighted sum of the number of new infections in the previous days.

distribution for negative effects and a uniform distribution for positive effects (Fig 3B). This prior allows for small increases in the number of new infections with a probability of 10%, while being uninformative about positive effects leading to reductions in the number of new infections.

In the second component, the expected number of new cases is calculated as a weighted sum of the number of new infections in the previous days. The weights reflect the distribution of the time from infection to reporting, and this distribution is estimated from the data assuming a log-normal distribution. Informative priors are used for the parameters of the log-normal

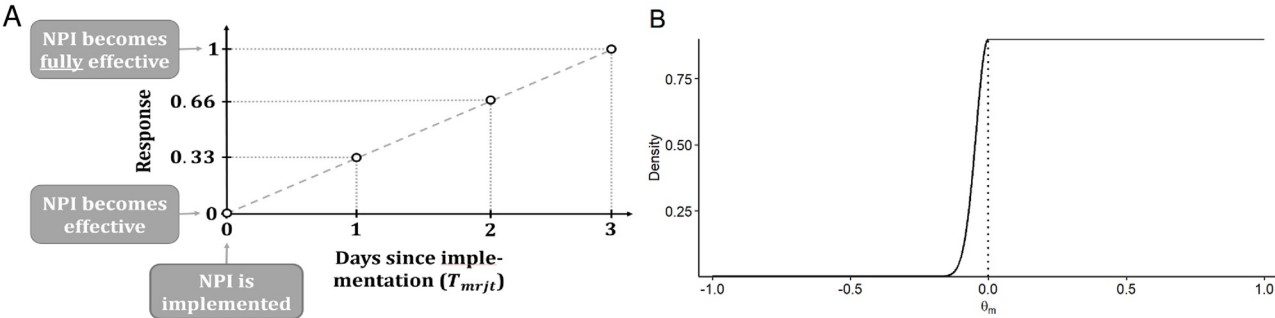

**Fig 3. Modeling choices for the effects of NPIs.** (A) Time-delayed response function as a first-order spline. (B) Prior for the effects of NPIs $\theta_m$.

distribution, reflecting prior knowledge. These priors are derived by decomposing the time from infection to reporting into the incubation period and the reporting delay, thereby using a meta-analysis [17] and estimates from multiple studies (in particular [18]) to obtain prior information about the distributions. The observed number of new cases is then modeled as a negative binomial distribution with the specified mean, allowing for overdispersion. Note that this part of the overall model was only applied after the day when a country reached 100 reported cumulative cases (called the modeling phase), in order to avoid modeling of highly irregular case numbers in the very early stages of the epidemic when most countries still had to set up their reporting practices. At the preceding days (called non-modeling phase), only the other two components of the model were used (i.e., the number of new infections and contagious subjects).

In the third component, the number of contagious subjects is calculated as a weighted sum of the number of new infections in the previous days. The weights reflect the probability of being contagious on a specific day after being infected and can be determined from the generation time distribution. This distribution is assumed to be known and our choice is based on an estimate by a recent study using data on the exposure for both the index and secondary case [19].

## 2.3 Simulation-based study

Highly parametrized models may raise concerns regarding the identifiability of individual model parameters. It is thus recommended to check if the true parameters can be recovered from the model using fake data simulation [20]. We performed such a simulation-based study, thereby demonstrating that it is possible to recover the true effect of NPIs within the uncertainty implied by the fitted posterior distribution of our model (Section 3 in S1 Appendix).

## 2.4 Parameter estimation

Choices of priors not mentioned so far are weakly informative, following general recommendations in Bayesian modeling [21]. All model parameters are estimated with a semi-mechanistic Bayesian hierarchical model. Specifically, Markov chain Monte Carlo (MCMC) sampling is used as implemented by the Hamiltonian Monte Carlo algorithm with the No-U-Turn Sampler (NUTS) from the probabilistic programming language Stan, version 2.19.2 [22]. Each model is estimated with 4 Markov chains and 2,000 iterations of which the first 1,000 iterations are discarded as part of the warm-up. Estimation power is evaluated via the ratio of the effective sample size ($\hat{n}_{\text{eff}}/N$), and convergence of the Markov chains is assessed with the Gelman-Rubin convergence diagnostic ($\hat{R}$). Further checks pertain to the detection of influential observations and correlations between the parameters of interest. If not stated otherwise, we report posterior means and credible intervals (CrIs) based on the 2.5% and 97.5% quantile of the posterior samples. Detailed estimation results and model diagnostics are provided in Section 5 in S1 Appendix.

The sensitivity of the results to the following modeling choices was investigated: start and end of the modeling phase, time-delayed response function, prior distribution for the effect of an NPI $\theta_m$, prior choices for the time from infection to reporting and for the generation time. Furthermore, the sensitivity of the results with respect to exclusion of individual countries was assessed with a leave-one-country-out analysis. Finally, a comparison to a similar study that was recently published [11] is presented.

## 2.5 Data and code availability

We collected data from publicly available data sources (Johns Hopkins Coronavirus Resource Center [16] for epidemiological data; news reports and government resources for policy measures). All the public health information that we used is documented in the main text, the extended data, and supplementary tables. A preprocessed data file together with reproducible code is available from https://github.com/nbanho/npi_effectiveness_first_wave.

# 3 Results

## 3.1 Estimated effects of NPIs

Using data from the first epidemic wave, we estimated the relative reduction in the number of new infections for each NPI (Fig 4A). Bans of large gatherings were associated with the highest reduction in the number of new infections (37%; 95% CrI 21% to 50%). The reduction was lower for venue closures (18%; 95% CrI −4% to 40%) and school closures (17%; 95% CrI −2% to 36%), followed by border closures (10%; 95% CrI −2% to 21%) and bans of small gatherings (9%; 95% CrI −4% to 23%). Stay-at-home orders (4%; 95% CrI −6% to 17%) and work-from-home orders (1%; 95% CrI −8% to 12%) appeared to be the least effective among the NPIs considered in this analysis.

The estimates for the individual effects suggest a particular strong effect of bans of large gatherings. This result is further supported by analyzing the posterior ranking of the effects (Fig 4B), which indicates that we could be at least 98% sure that bans of large gatherings were

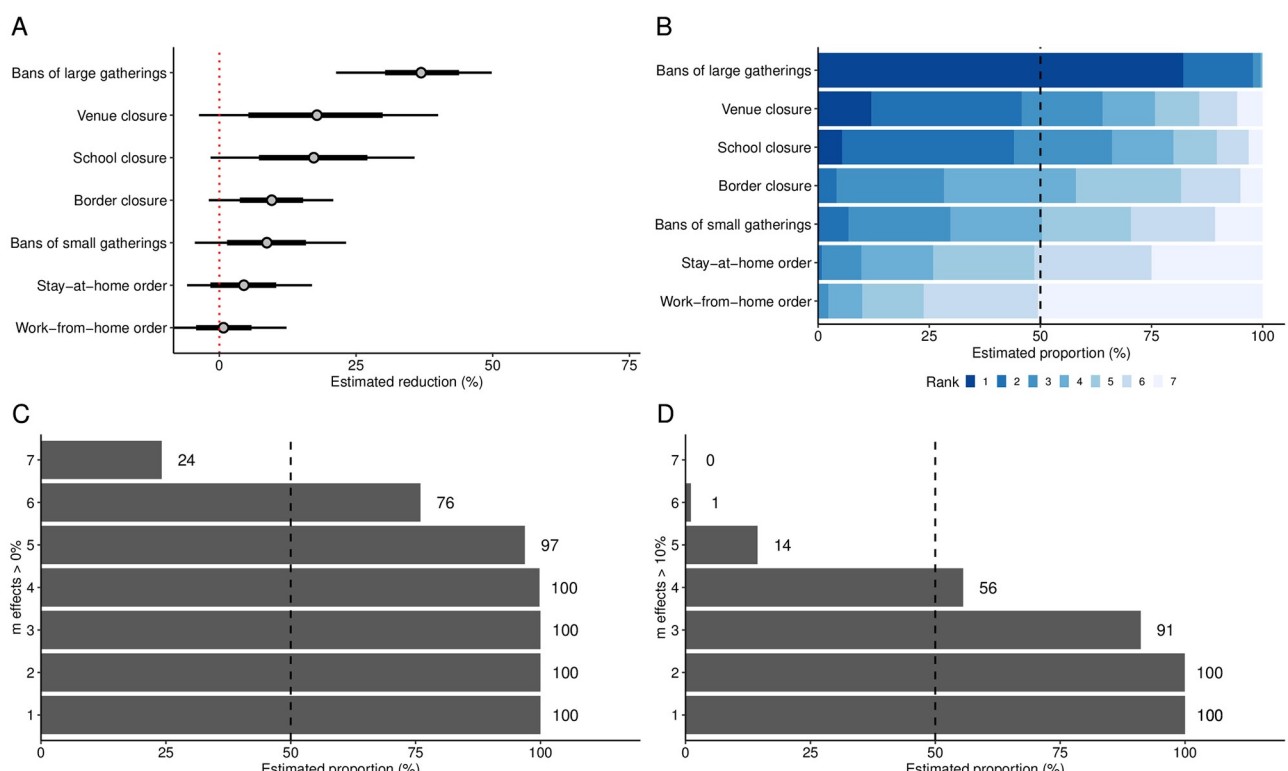

**Fig 4. Estimated effects of NPIs.** (A) Reduction in new infections (posterior mean as dots with 80% and 95% credible interval as thick and thin lines, respectively). (B) Ranking of the effects of NPIs from highest (1) to lowest (7) (posterior frequency distribution). (C) Frequency of at least *m* positive effects (posterior frequency distribution). (D) Frequency of at least *m* effects greater than 10% (posterior frequency distribution).

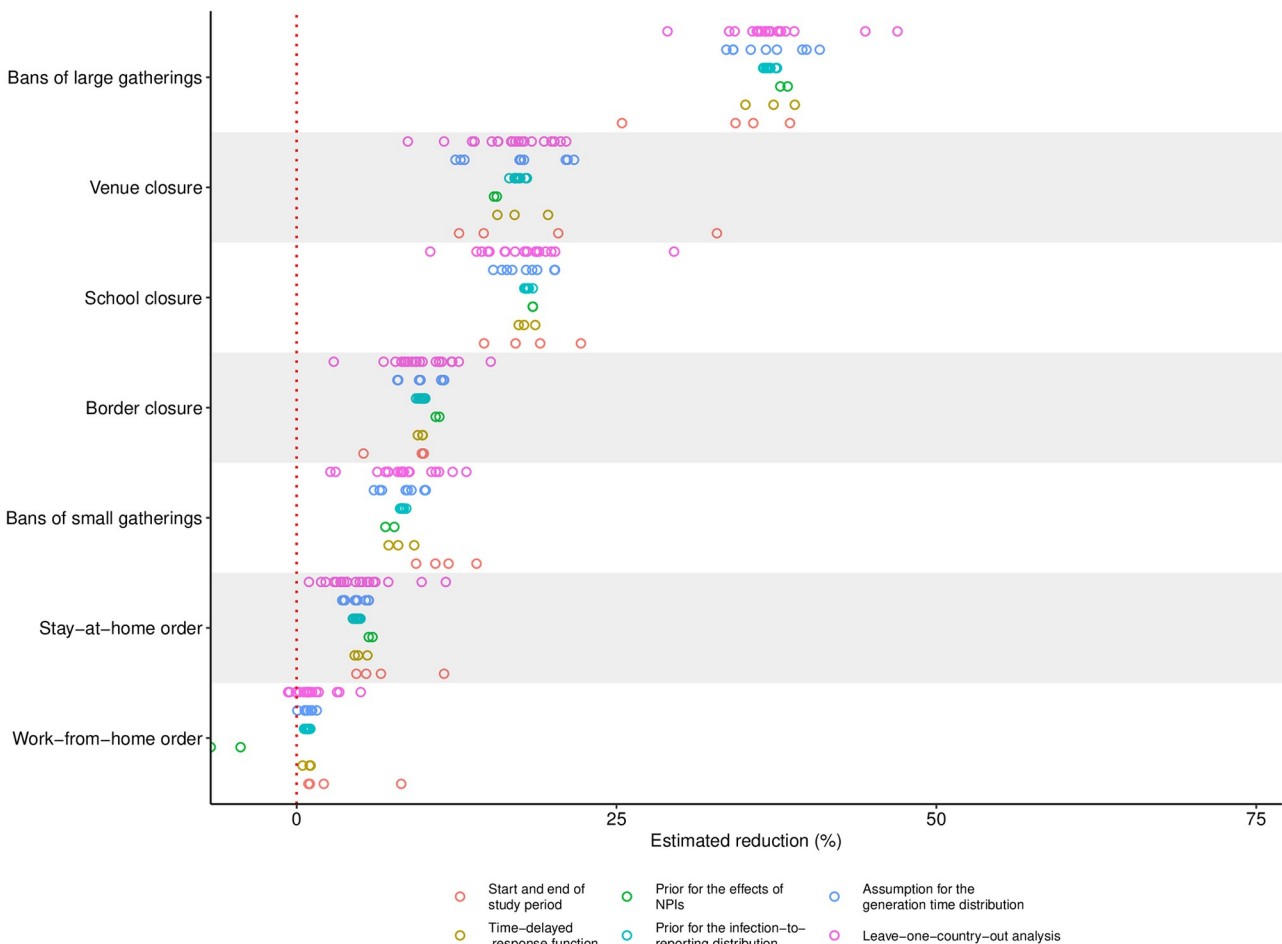

**Fig 5. Summary of the sensitivity analysis.** Sensitivity of the estimated effects of NPIs (posterior mean as dots) to different data preprocessing, varying modeling and prior choices, and data exclusion. Section 6 in S1 Appendix presents all individual sensitivity analyses in detail.

among the two most effective NPIs. Conversely, we could be at least 76% sure that work-from-home orders were among the two least effective NPIs.

All NPIs together lead to an estimated relative reduction in the number of new infections by 67% (95% CrI 64% to 71%). The combined effectiveness of NPIs was also analyzed (Fig 4A–4D). Thereby, we could be at least 97% sure that there are five NPIs, which each lead to a reduction in the number of new infections. Regarding the magnitude of the effects, we could be at least 91% sure that there are three NPIs, which each lead to a reduction in the number of new infections of more than 10%.

### 3.2 Sensitivity analysis

Sensitivity of our results was assessed with respect to small variations in the start and end of the study period, alternative choices for the time-delayed response function, the prior distribution for the effects of NPIs, the prior distribution for the delay distributions, and a leave-one-country-out analysis (Fig 5). Overall, our results were only slightly sensitive to alternative modeling and prior choices, particularly when deriving conclusions about the ranking of the effects. The leave-one-out analysis for the countries in our sample indicated some sensitivity

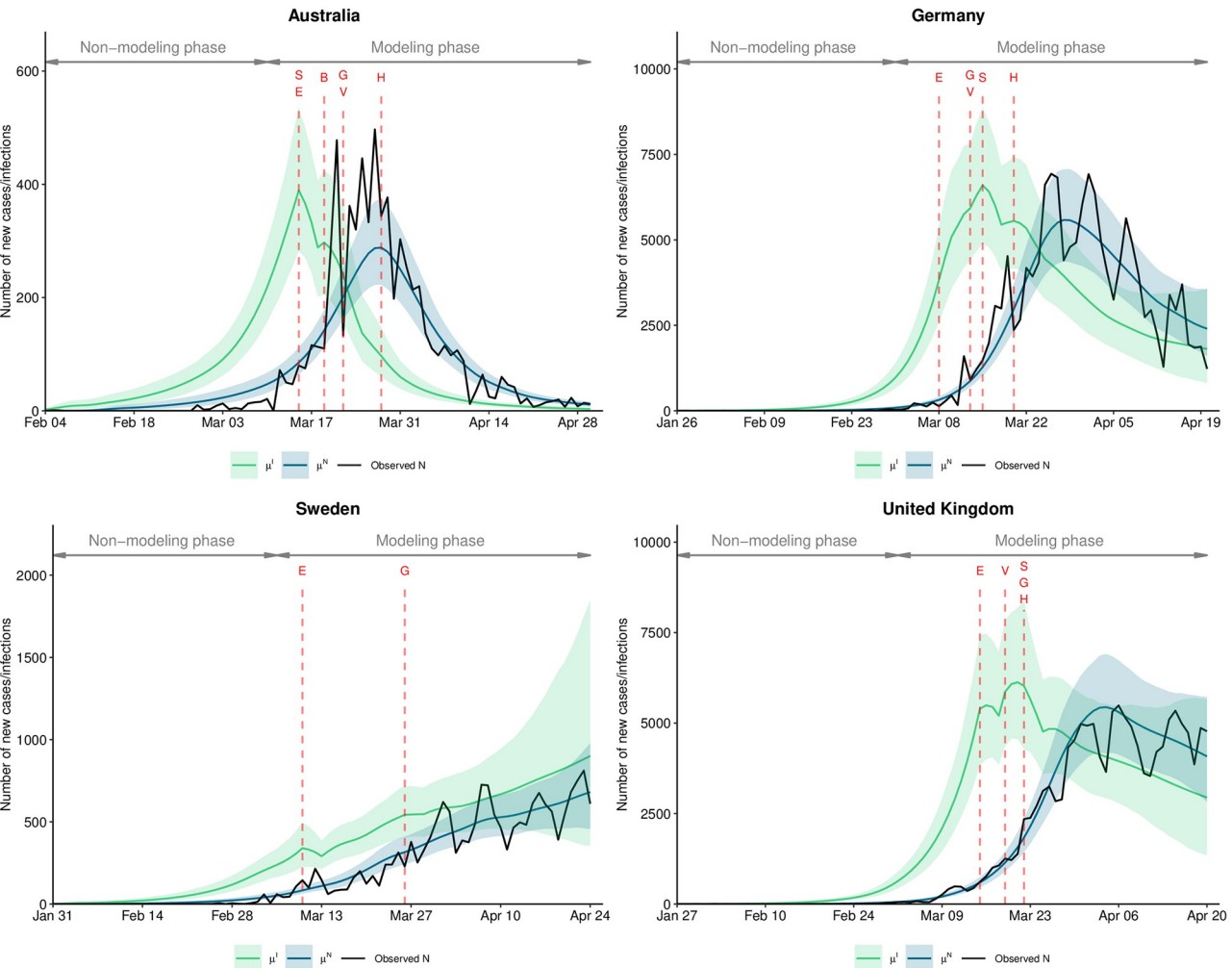

**Fig 6. Model fit for four selected countries over time.** Expected number of new infections $\mu^I$ and new cases $\mu^N$ (posterior mean as colored lines with 95% credible interval as shaded area) and the observed number of new cases by country over time. Red letters and lines indicate the first day an NPI was implemented within a country (S: School closures, B: Border closure, E: Ban of large gatherings, G: Ban of small gatherings, V: Venue closure, H: stay-at-home order, W: Work-from-home order). The non-modeling phase is the time period before 100 cumulative cases were observed, which was used to seed infections in the early outbreak of the epidemic. Plots for all countries are provided in Section 7 in S1 Appendix.

with respect to the exclusion of Australia, Sweden, and Switzerland. This sensitivity was further investigated and subsequent analysis indicated that data from these countries are particularly informative for the effect of school closures and bans of large gatherings.

### 3.3 Estimated number of new infections and cases over time

The model fit was assessed by comparing the expected number of new cases based on our model to the observed number of new cases (Fig 6). Here an acceptable degree of agreement is observed for Australia, Germany, Sweden, the UK, and all other countries (figures for all other countries are listed in Section 7 in S1 Appendix). In addition, the posterior distribution for the number of new infections is depicted, illustrating how NPIs lead to a reduction in new infections, which later implies a reduction in new cases.

### 3.4 Comparison with a similar study

Recently, an analysis similar to our was published by Brauner et al. [11]. In the following, we present a brief comparison. Brauner et al. used the same data source to generate country-specific case numbers per day, but included 30 countries only partially overlapping with our selection. The authors used similar sources to collect data on NPIs but classified some NPIs differently. The modeling approach was very similar, except for avoiding to take the number of contagious subjects explicitly into account and considering both the number of reported cases and reported deaths as outcomes. The authors included country-specific NPI effects (although not explicitly analyzed), while we took into account the share of the country's total population being affected by NPIs and a time-delayed response to their implementation. The authors used similar priors for the distribution of the time from infection to reporting and for the generation time distribution, but, in contrast to our work, Brauner et al. estimated the latter explicitly from data as part of fitting the model.

To facilitate a comparison of the results between the modeling approach by Brauner et al. and our approach, we reanalyzed their data using our model. The estimated effects from our model were generally smaller but the overall ranking of the effects was very similar when fitted to the data by Brauner et al. (Table 2A). When applying both models to our data, the overall ranking is still similar but with small differences in the estimates for some NPIs, in particular a higher estimate for the effect of school closures and a lower estimate for the effect of bans of small gatherings as compared to the model by Brauner et al. (Table 2C). In contrast to that, when comparing the results from our model on different datasets (our data and the data from Brauner et al.), there was only a similarity in the overall pattern (Table 2C). For instance, we found similar strong effects for bans of large gatherings and school closures as well as weak effects for stay-at-home orders, but the effect of venue closures is stronger and the effect of work-from-home orders weaker in our data. In addition, the larger set of countries in the data from Brauner et al. was associated with smaller credible intervals for the estimates.

## 4 Discussion

### 4.1 Estimated NPI effects

We performed a cross-country analysis based on $n = 20$ countries in order to assess the effectiveness of seven NPIs during the first epidemic wave. Our findings suggest that some NPIs, particularly in combination, lead to a strong reduction in the number of new infections. Among the NPIs considered, bans of large gatherings were most effective in reducing the number of new infections across countries. Bans of large gatherings are targeted towards large gatherings of people and may thus prevent so-called "superspreader events", which have been shown to account for a substantial fraction of the total number of infections [23–25]. Many superspreading events originated in points-of-interest such as bars and restaurants [26], which aligns with our finding of a sizeable reduction from venue closures.

The effectiveness of school closures in transmission control is subject to debate [27, 28]. Although children are less susceptible to the virus than adults, it is less clear how children and adults compare regarding their infectiousness [29]. Early findings suggested that school closures were only marginally effective in transmission control [30]. Our results provide contrary evidence, in line with recent findings from other population-based studies [11, 13, 15, 31, 32]. However, note that our findings relate to the closure of primary schools, which often coincided with the closure of secondary schools and universities. The study by Brauner et al. differentiated between NPIs for the closure of schools and the closure of universities, but could afterwards not disentangle the estimated effects [11]. Thus, it is subject to further investigation

**Table 2. Comparison of modeling and data with results from Brauner et al. [11].** (A) Estimated effects by model and data from Brauner et al. (posterior mean in%, 95% prediction interval (PrI) and rank) and by our model and data from Brauner et al. (posterior mean in%, 95% credible interval and rank). Note that, in these analyses, we report cumulative effects for bans of small gatherings and businesses closed as in Brauner et al. (B) Estimated effects on our data by the model from Brauner et al. (posterior mean in%, 95% prediction interval (PrI) and rank) and by our model (posterior mean in%, 95% credible interval, and rank). (C) Estimated effects by our model on our data (posterior mean in%, 95% credible interval, and rank) and by our model and data from Brauner et al. (posterior mean in%, 95% credible interval, and rank). Similar NPIs were matched but their definitions are not exactly the same. Note that in applying our model to the data by Brauner at al., we report the cumulative effect of "Gatherings <1000" and "Gatherings <100" when referring to our ban of small gatherings.

(A) Data by Brauner et al. but different modeling.

|  | Model and data by Brauner et al | | Our model and data by Brauner et al | |
|---|---|---|---|---|
|  | Mean (PrI) | Rank | Mean (CrI) | Rank |
| Gatherings <1000 | 23 (0—40) | 5 | 17 (5—28) | 5 |
| Gatherings <100 | 34 (12—42) | 3 | 24 (11—36) | 2 |
| Gatherings <10 | 42 (17—60) | 1 | 32 (17—47) | 1 |
| Schools and universities closed | 38 (16—54) | 2 | 23 (12—34) | 3 |
| Some businesses closed | 18 (− 8—40) | 6 | 5 (− 5—17) | 7 |
| Most businesses closed | 27 (− 3—49) | 4 | 18 (5—30) | 4 |
| Stay-at-home order | 13 (− 5—31) | 7 | 12 (5—18) | 6 |

(B) Our data but different modeling.

|  | Model by Brauner et al and our data | | Our model and our data | |
|---|---|---|---|---|
|  | Mean (PrI) | Rank | Mean (CrI) | Rank |
| Ban of large gatherings | 48 (26—63) | 1 | 37 (21—50) | 1 |
| Venue closure | 30 (0—57) | 2 | 18 (4—40) | 2 |
| School closure | 7 (13—31) | 4 | 17 (2—36) | 3 |
| Border closure | 6 (−12—26) | 5 | 10 (2—21) | 4 |
| Ban of small gatherings | 15 (− 6—38) | 3 | 9 (− 4—23) | 5 |
| Stay-at-home order | 5 (−13—24) | 7 | 4 (6—17) | 6 |
| Work-from-home order | 6 (−12—27) | 6 | 1 (8—12) | 7 |

(C) Our model but different data.

| Our NPI | NPI by Brauner et al. | Our model and our data | | Our model and data by Brauner et al. | |
|---|---|---|---|---|---|
|  |  | Mean (CrI) | Rank | Mean (CrI) | Rank |
| Ban of large gatherings | Gatherings <1000 + Gatherings <100 | 37 (21—50) | 1 | 24 (11—36) | 1 |
| Venue closure | Some businesses closed | 18 (− 4—40) | 2 | 5 (− 5—17) | 6 |
| School closure | Schools and universities closed | 17 (− 2—36) | 3 | 23 (12—34) | 2 |
| Border closure |  | 10 (− 2—21) | 4 |  |  |
| Ban of small gatherings | Gatherings <10 | 9 (− 4—23) | 5 | 9 (2—19) | 5 |
| Stay-at-home order | Stay-at-home order | 4 (− 6—17) | 6 | 12 (5—18) | 4 |
| Work-from-home order | Most businesses closed | 1 (− 8—12) | 7 | 12 (2—22) | 3 |

whether the closure of primary schools is less effective than the closure of secondary schools and universities [29].

A small effect was estimated for stay-at-home orders. This seems to contradict findings about the high effectiveness of the lockdown from Flaxman et al. [12]. However, one should consider that their definition of a lockdown encompasses multiple NPIs that we differentiated (e.g., bans of small gatherings, venue closures, and stay-at-home orders). Taken together, our estimated "lockdown effect" would therefore also be large. Finally, although our findings suggest that work-from-home orders were not effective, it should be considered that our definition of a work-from-home order referred to a mandatory order to work from home for non-essential business activities, while many countries only issued recommendations.

## 4.2 Methodological aspects

Flaxman et al. [12] were the first who attempted to link NPIs to observed cases or deaths using a semi-mechanistic Bayesian hierarchical model. Both the study by Brauner et al. [11] and our study can be seen as extensions of this approach, thereby making use in particular of data from more countries. This also implies the possibility of refined modeling. For instance, Flaxman et al. makes explicit assumptions on the distribution of the time from infection-to-case confirmation, which is estimated from data in the study of Brauner et al. and our study. Furthermore, modeling of the NPI effects was refined by considering country-specific effects in Brauner et al. and by taking into account regional variation in the implementation of NPIs in our study. We discuss methodological aspects in comparison to Flaxman et al. and Brauner et al. in more detail in Section 2 in S1 Appendix.

## 4.3 Insights from a comparison with a similar study

With the data on reported cases and deaths gathered by the John Hopkins University and data on NPIs gathered by government and news websites, there is a unique source for analyzing the impact of health policy measures on the course of the COVID-19 pandemic. However, there are many degrees of freedom with respect to preprocessing the available data, constructing a model, and making prior assumptions and choices. To understand the influence of these data and modeling choices on the results, it is desirable that different research groups analyse the same data in different ways. A very recent publication by Brauner et al. [11] gave us the possibility to perform a first check of this type. The check indicated that the choice of countries and definition of NPIs has a larger influence on the estimated effects than the detailed choices in modeling. A re-analysis of the original study by Flaxman et al. could also demonstrate sensitivity of the results with respect to specific choices of the definition of NPIs, but also found sensitivity to specific modeling choices [33].

## 4.4 Limitations

Our analysis is subject to limitations. First, our modeling assumptions do not allow for (random) variation in the effect of NPIs across countries and assume a fixed effect. Brauner et al. [11] demonstrated that, in principle, it is feasible to allow for country-specific variation in the effects. However, the specific parametrization chosen in our model (i.e., to take into account regional variation in the implementation of NPIs via the population share) made it challenging to incorporate random variation of the NPI effect $\theta_m$ across countries. Brauner et al observed in the sensitivity analysis of their study that assuming a fixed effect does not alter the main conclusions that they discussed regarding the effectiveness of NPIs.

Second, any approach of explaining changes in the observed number of cases solely by specific NPIs makes the implicit assumption that these changes were not the result of some other factors. For instance, it is possible that additional measures or an increasing general awareness encouraged social distancing and hence lead to less infections. If this is the case, such effects will erroneously be assigned to the NPIs and possibly overstate their overall impact.

Third, it is challenging to distinguish between the effects of single NPIs due to their concurring introduction in many countries (Table 4 in S1 Appendix). This is reflected by wide credible intervals and a negative association between effects (Section 5.2 in S1 Appendix), suggesting that the effect of one NPI may be attributed partially to another. Note that the effect of bans of large gatherings could be estimated with comparably narrower credible intervals than that of venue and school closures. A reason for this could be that all three measures were implemented by most countries but bans of large gatherings had on average a greater distance in implementation to other NPIs as compared to school and venue closures (Table 3 in S1

Appendix). Despite the difficulty in disentangling the effects of individual NPIs, we were able to demonstrate that we can be fairly confident with respect to their relative ranking.

Fourth, our analysis is limited by the type of data utilized to define the outcomes. Using the number of reported cases as outcome implies that reporting practices may have an influence on the results. In particular, definitions and reporting practices differed between countries and over time. However, since we modeled the ratio between the number of new cases and the number of contagious subjects, we can expect that these effects cancel out. The same argument applies to the challenge that we have to expect a substantial number of undetected cases, with detection rates varying across countries and over time. In addition, we modelled the number of reported cases as a random variable, thereby allowing for overdispersion. Moreover, we took into account that countries had still to develop their reporting practice in the very early phase of the epidemic by starting the modeling phase after at least 100 cumulative cases were reported in a country.

Fifth, it is not considered in our analysis that the number of susceptible people in the population decreases as the number of people that were already infected increases over time. However, we think this limitation is not of particular concern during the first epidemic wave. A study in Spain, which experienced a severe first epidemic wave, showed that prevalence was only around five percent in the population, indicating that the large majority of the Spanish population was still susceptible after the first epidemic wave [34]

Sixth, our analysis is limited by the need to classify NPIs in a comparable manner across countries and to determine dates when NPIs were exactly implemented. There is always some subjectivity in making corresponding decisions. The comparison of our study with the one by Brauner et al. suggests that such definitions can be crucial. We provide a detailed description of our decisions (Section 8 in S1 Appendix).

## 4.5 Outlook

With the model presented in this paper and that developed by Brauner et al., we may have reached a first level of maturity. This may be sufficient to justify the use of these models in analyzing the effects of NPIs. In principle, these models could also be used to study the effect of lifting NPIs. A natural next step would be to apply these models to data from the second wave, which is still ongoing. An interesting question would then be whether we can expect similar effects in the first and the second wave. It is likely that some effects may have changed, as the situations are not necessarily comparable and experience from the first wave may have helped in dealing with the second wave.

Of course, there is still room for improvement and refinement of the models. If information on new cases is also available at a regional level, regional variation in implementing NPIs can be used as an additional source of information using a two-level hierarchical approach. Furthermore, between-country and regional variation may be linked to certain characteristics which may help to understand the conditions under which an NPI is most effective. Similarly, information on cases stratified by patient characteristics may provide new insights. Also the modeling of the outcome can be improved, e.g., by taking weekday effects into account.

## 5 Conclusion

Our analysis makes a contribution to the emerging evidence about the effectiveness of different NPIs in the first epidemic wave. Ideally, such studies could inform public health policy in the onset of future epidemics, as well as modeling efforts related to later waves. In our study, bans of large gatherings in particular were identified as an effective measure, whereas little evidence was found for substantial effects of stay-at-home orders and work-from-home orders. A

comparison with another study indicates robustness of some conclusions, but also a dependence on the choice of data and definitions.

## Supporting information

**S1 Appendix. Supplementary material.** The supplementary material contains (1) a detailed description of the method, (2) discussion of methodological aspects in comparison to related work, (3) description and results from a simulation-based study, (4) further descriptives on the timing of non-pharmaceutical interventions, (5) detailed estimation results and model checks, (6) results from the sensitivity analysis, (7) a visual inspection of the model fit, and (8) data on non-pharmaceutical interventions.
(PDF)

## Acknowledgments

We thank various people around the world for checking our data on non-pharmaceutical interventions.

## Author Contributions

**Conceptualization:** Nicolas Banholzer, Stefan Feuerriegel, Werner Vach.

**Data curation:** Nicolas Banholzer.

**Formal analysis:** Nicolas Banholzer, Werner Vach.

**Funding acquisition:** Stefan Feuerriegel.

**Investigation:** Nicolas Banholzer, Eva van Weenen, Adrian Lison, Alberto Cenedese, Arne Seeliger, Bernhard Kratzwald, Daniel Tschernutter, Joan Puig Salles, Pierluigi Bottrighi, Werner Vach.

**Methodology:** Nicolas Banholzer, Sonja Lehtinen, Werner Vach.

**Project administration:** Nicolas Banholzer, Stefan Feuerriegel, Werner Vach.

**Resources:** Stefan Feuerriegel.

**Software:** Nicolas Banholzer.

**Supervision:** Stefan Feuerriegel, Werner Vach.

**Validation:** Nicolas Banholzer.

**Visualization:** Nicolas Banholzer, Werner Vach.

**Writing – original draft:** Nicolas Banholzer, Stefan Feuerriegel, Werner Vach.

**Writing – review & editing:** Nicolas Banholzer, Eva van Weenen, Adrian Lison, Arne Seeliger, Bernhard Kratzwald, Daniel Tschernutter, Sonja Lehtinen, Stefan Feuerriegel, Werner Vach.

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
