## [Decision Letter · Decision Letter 0]

23 Apr 2021

PONE-D-21-01565

Estimating the effects of non-pharmaceutical interventions on the number of new infections with COVID-19 during the first epidemic wave

PLOS ONE

Dear Dr. Banholzer,

Thank you for submitting your manuscript to PLOS ONE. After careful consideration, we feel that it has merit but does not fully meet PLOS ONE’s publication criteria as it currently stands. Therefore, we invite you to submit a revised version of the manuscript that addresses the points raised during the review process.

First of all I apologize for the delay in peer-review process, but even if COVID-19 studies are prioritized, the resources are still limited in terms of finding available reviewers due to the large number of manuscript submitted on this topic. Regarding the manuscript, please note that the two reviewers came to diverging conclusions (minor revisions vs. reject). The paper originality is questioned by Reviewer #2, i.e. no significant advancement with respect to Brauner et al. study. Moreover, also some methodological aspects should be addressed. However, as recognized by Reviewer #2, the study is interesting and for this reason I reached the "Major Revision" decision. The authors should address and respond point-by-point to the reviewer issues/comments when drafting the new version.

We look forward to receiving your revised manuscript.

Kind regards,

Simone Lolli

Academic Editor

PLOS ONE

Additional Editor Comments:

First of all I apologize for the delay in peer-review process, but even if COVID-19 studies are prioritized, the resources are still limited in terms of finding available reviewers due to the large number of manuscript submitted on this topic.

Regarding the manuscript, please note that the two reviewers came to diverging conclusions (minor revisions vs. reject). The paper originality is questioned by Reviewer #2, i.e. no significant advancement with respect to Brauner et al. study. Moreover, also some methodological aspects should be addressed. However, as recognized by Reviewer #2, the study is interesting and for this reason I reached the "Major Revision" decision. The authors should address and respond point-by-point to the reviewer issues/comments when drafting the new version.

Journal Requirements:

[JPS declares part-time employment at Luciole Medical outside of the submitted work.].

3. We note you have included a table to which you do not refer in the text of your manuscript. Please ensure that you refer to Table 1 and 2 in your text; if accepted, production will need this reference to link the reader to the Table.

Reviewers' comments:

Reviewer's Responses to Questions

**Comments to the Author**

1. Is the manuscript technically sound, and do the data support the conclusions?

Reviewer #1: Yes

Reviewer #2: Partly

2. Has the statistical analysis been performed appropriately and rigorously? 

Reviewer #1: Yes

Reviewer #2: Yes

3. Have the authors made all data underlying the findings in their manuscript fully available?

Reviewer #1: Yes

Reviewer #2: Yes

4. Is the manuscript presented in an intelligible fashion and written in standard English?

Reviewer #1: Yes

Reviewer #2: Yes

5. Review Comments to the Author

Reviewer #1: Please see attached pdf. I will now add random letters to fill the required character count. sdoinev9ghureiabnaoin498nnv9p2q4meomug pr98qut qp3 9 iug89 983u qjoizn389 4u3g8imajmguieu 983 jivgzmh3pjiow 9w845w gmjeg 3p.

Reviewer #2: Review comments for the manuscript, “Estimating the effects of non-pharmaceutical interventions on the number of new infections with COVID-19 during the first epidemic wave” by Banholzer et al.

• A semi-mechanistic Bayesian hierarchical model to assess the effectiveness of seven NPIs in curtailing the number of COVID-19 cases in 20 countries using data for first wave is proposed. This is an interesting study, but in addition to the comments below, I believe that it does not provide significant advancement in the field beyond that in the study in Brauner et al. mentioned in the paper.

• No motivation on why using data for the first pandemic wave at a time when many countries have or are already experiencing second or third waves is appropriate has been provided. Hence, it is not clear how useful this study is in the current fight against the pandemic, especially since most of the measures considered in this study were relaxed or completely discontinued months ago in most of the study countries.

• Use of the phrase NPIs in this manuscript is misleading. NPIs are broadly defined to include social-distancing, lockdowns, wearing of masks in public, contact-tracing, quarantine, isolation, hand-washing, etc. I believe the seven measures in the study fall under one or two broad kinds of NPIs, e.g., social-distancing and lockdowns, especially since most of them overlap. Strangely, the manuscript is silent on how these overlaps between the selected control measures were handled or eliminated.

• The authors say, “On the one hand, the national strategies consisted of similar NPIs, which we can expect to work in a similar manner, despite cultural and organisational differences between countries.” This is not true, since not all the countries, e.g., the US, implemented the identified strategies nationally. Different states in the US implemented different strategies with different stringency levels. I also think it is not reasonable not to consider some of the countries that have been able to manage the pandemic successfully through the use of NPIs in this study.

• I am not sure I understand what is going on from Figure 2, especially since the flow branch 1 has no direction.

• It is strange that the new infections as modeled in Eq. 1 are generated only from the transmission rate, and the number of contagious subjects, but not also the uninfected. Also, introducing NPIs as in this equations leads to unidentifiability, where different inputs can lead to the same output. How was this issue handled?

6. PLOS authors have the option to publish the peer review history of their article (what does this mean?). If published, this will include your full peer review and any attached files.

Reviewer #1: **Yes: **Jan M. Brauner

Reviewer #2: No

---

## [Author Response · Author response to Decision Letter 0]

14 May 2021

See File Response to the Reviewers.

---

## [Decision Letter · Decision Letter 1]

24 May 2021

Estimating the effects of non-pharmaceutical interventions on the number of new infections with COVID-19 during the first epidemic wave

PONE-D-21-01565R1

Dear Dr. Banholzer,

We’re pleased to inform you that your manuscript has been judged scientifically suitable for publication and will be formally accepted for publication once it meets all outstanding technical requirements.

Kind regards,

Simone Lolli

Academic Editor

PLOS ONE

Additional Editor Comments (optional):

The manuscript is now ready for publication.

Reviewers' comments:

Reviewer's Responses to Questions

**Comments to the Author**

1. If the authors have adequately addressed your comments raised in a previous round of review and you feel that this manuscript is now acceptable for publication, you may indicate that here to bypass the “Comments to the Author” section, enter your conflict of interest statement in the “Confidential to Editor” section, and submit your "Accept" recommendation.

Reviewer #1: All comments have been addressed

2. Is the manuscript technically sound, and do the data support the conclusions?

Reviewer #1: Yes

3. Has the statistical analysis been performed appropriately and rigorously? 

Reviewer #1: Yes

4. Have the authors made all data underlying the findings in their manuscript fully available?

Reviewer #1: Yes

5. Is the manuscript presented in an intelligible fashion and written in standard English?

Reviewer #1: Yes

6. Review Comments to the Author

Reviewer #1: All my comments have been addressed. Congratulations to the authors for the nice paper.

7. PLOS authors have the option to publish the peer review history of their article (what does this mean?). If published, this will include your full peer review and any attached files.

Reviewer #1: **Yes: **Jan Brauner

---

## [Editor Report · Acceptance letter]

26 May 2021

PONE-D-21-01565R1 

Estimating the effects of non-pharmaceutical interventions on the number of new infections with COVID-19 during the first epidemic wave 

Dear Dr. Banholzer:

I'm pleased to inform you that your manuscript has been deemed suitable for publication in PLOS ONE. Congratulations! Your manuscript is now with our production department. 

Kind regards, 

on behalf of

Dr. Simone Lolli 

Academic Editor

PLOS ONE